# Binary Mixtures of Essential Oils: Potent Housefly Adulticides That Are Safe Against Non-Target Species

**DOI:** 10.3390/insects16080855

**Published:** 2025-08-17

**Authors:** Hataichanok Passara, Sirawut Sittichok, Tanapoom Moungthipmalai, Chamroon Laosinwattana, Kouhei Murata, Mayura Soonwera

**Affiliations:** 1Office of Administrative Interdisciplinary Program on Agricultural Technology, School of Agricultural Technology, King Mongkut’s Institute of Technology Ladkrabang, Ladkrabang, Bangkok 10520, Thailand; hataichanok.pa@kmitl.ac.th (H.P.); 64604012@kmitl.ac.th (T.M.); chamroon.la@kmitl.ac.th (C.L.); 2School of Agriculture and Cooperatives, Sukhothai Thammathirat Open University, Nonthaburi 11120, Thailand; sirawut.sit@stou.ac.th; 3School of Agriculture, Tokai University, Kumamoto 862-8652, Japan; kmurata@agri.u-tokai.ac.jp

**Keywords:** adulticide, binary mixture, essential oils, *Eucalyptus globulus*, geranial, *Musca domestica*, non-target species

## Abstract

One of the most universally disruptive vectors of diseases mechanically transmitted to farm animals and humans is the housefly (*Musca domestica* L.). Fly control has been progressively complicated as more fly populations have become resistant to commonly used synthetic pesticides. Natural adulticides from essential oils and their main constituents from plants are potent compounds for controlling fly populations and corresponding diseases, such as anthrax, shigellosis, cholera, and coronavirus diseases. In the present study, we examined the fly adulticidal efficacy and reliability of mixtures of geranial, eucalyptol, *Cymbopogon citratus* Stapf, and *Eucalyptus globulus* Labill. essential oils (EOs). Distilled water served as a negative control, whereas α-cypermethrin served as a positive control. The 2.5% (*v*/*v*) geranial + 2.5% (*v*/*v*) *E. globulus* EO mixture was the most effective and long-acting. Most significantly, all the individual and mixture EO formulas were harmless to a non-target pollinator (*Apis florea* Fabricius) and a common aquatic predator (*Poecilia reticulata* Peters); in comparison, *α*-cypermethrin was highly toxic and caused significant harm. This mixture of geranial and *E. globulus* EO formula is environmentally friendly. It should be further developed into a commercial adulticide for fly population control to reduce the prevalence of diseases vectored by houseflies.

## 1. Introduction

Houseflies (*Musca domestica* L.) are leading pests in cattle farming, medicine, and human health [1,2], transmitting more than 100 diseases and infections, including typhoid fever, food poisoning, eye infections, parasitic infections, and respiratory and reproductive infections in cattle [1,3]. Flies also spread bacteria such as *Salmonella*, *Shigella*, and *Giardia lamblia*, in addition to protozoa and trophozoites [4]. They contaminate water and food, impacting livestock performance due to consequent poor nutrition, decreased body weight, and reduced milk supply [3]. In 2021, the U.S. agricultural sector suffered losses of more than USD 1 billion due to houseflies [2,5,6]. Managing fly populations is challenging, particularly in infested areas [6]. Synthetic insecticides are commonly employed in such areas; flies can develop resistance to these measures, however, and insecticides can harm the environment, non-target species, and humans [7,8,9,10,11,12]. Natural insecticides, such as essential oils (EOs) and their primary chemical constituents, thus represent safer alternatives to synthetic insecticides [13,14,15].

Plant-derived EOs have a lesser impact on mammals and non-target species [16,17,18]. Plant EOs are suitable for sustainable fly and mosquito management, showing low resistance development, rapid biodegradation, and environmental safety [19,20]. In key outbreak areas such as pre-kindergartens, daycare centers, nursing homes, restaurants, post-harvest facilities, and milkeries, EOs are a viable option for controlling fly populations [21,22,23]. EOs function as adulticides, pupicides, larvicides, ovicides, and oviposition deterrents against flies [24,25,26,27]. Plant EOs and their primary chemical constituents have potent adulticidal efficacy; however, they are safe for both humans and non-target species [28].

Essential oils (EOs) and their primary chemical components have been employed in a number of studies, with the obtained results demonstrating their effective elimination of adult flies. Several examples and their corresponding references are listed in Table 1.

Some geranial and *E. globulus* EO combined formulas possess potent adulticidal activity and have been reported to be safe for different ecosystems, non-target organisms, and humans [26,33]. In this study, we chose dwarf honeybees, *A. florea*, and guppies, *P. reticulata*, as the non-target organisms. In addition to being effective pollinators, honeybees perform several ecological services [34]. These honeybees and stingless bees stand as an efficient indicator of the level of insecticide residue in the environment [35,36]. Insecticides are also harmful to aquatic life in marine environments. In this study, we investigated a non-target aquatic organism, referred to as guppies, *P. reticulata*. These non-target organisms may serve as bioindicators to assess the level of toxicity within an ecosystem [37]. In contrast with *⍺*-cypermethrin, a common synthetic adulticide, these combined essential oils and main chemical constituent formulas are demonstrated herein to be an effective source of potent housefly adulticides that are relatively harmless to non-target species.

## 2. Materials and Methods

### 2.1. Essential Oils and Other Materials

*C. citratus* EO (CAS-No: 84238-19-7) and *E. globulus* EO (CAS-No: 8000-48-4) were acquired from Chemipan Corporation Co., Ltd., Bangkok, Thailand. Sigma-Aldrich Company Ltd. (Saint Louis, MO, USA) supplied the technical quality 98% geranial (CAS-No: 5392-40-5, the main ingredient of *C. citratus* EO) and 99% eucalyptol (CAS-No: 470-82-6, the primary compound of *E. globulus* EO) and identified the chemical structure formulas of the primary compounds (Figure 1). Every combination of EOs used in this investigation is listed in Table 2. Siribuncha Company Limited, Phra Khanong, Bangkok, Thailand, provided the 70% (*v*/*v*) ethyl alcohol stock solution used to produce the combination formulas. Distilled water was obtained from King Mongkut’s Institute of Technology, Ladkrabang (KMITL), Bangkok, Thailand’s School of Food Industry, and served as the negative control.

### 2.2. Fly Management

Adult *M. domestica* flies were collected using a sweep net from the Nong Chok market in Bangkok, Thailand. At the School of Agricultural Technology, King Mongkut’s Institute of Technology, Ladkrabang (KMITL), Bangkok, Thailand (latitude: 13°51′20′′ N, longitude: 100°51′45′′ E), an entomologist identified the obtained flies. Twenty males and females were subsequently raised in a 30 × 30 × 30 cm^3^ insect cage at room temperature (25.3 ± 2.5 °C and 75.2 ± 3.47% relative humidity). Adult flies were fed a 5:5:90 mixture of 10% glucose solution, milk powder, and mineral water. After two to three days of feeding, the female adults laid their eggs on steaming mackerel. On the fifth day, these eggs hatched into larvae in their first instar, which were subsequently raised on steamed mackerel until they became pupae. The pupae were gathered in a 15 × 15 × 15 cm^3^ plastic box lined with sterilized coconut husk. The pupae matured into adult flies within five days. Tests were conducted on the third day of the subject’s mature life.

### 2.3. Flies’ Adulticidal Activities

The World Health Organization’s (WHO) standard susceptibility test [38] was employed to determine the adulticidal activity against adult flies. The test was conducted in two test tubes of 4.4 cm in diameter and 12.5 cm in length. The first tube served as the treatment specimen, and the second tube initially contained a piece of intact filter paper. Ten 3-day-old adult flies (with a sex ratio of 1:1) were exposed to two milliliters of each formula, as detailed in Table 2. Each treatment formula was dropped onto a 12-by-15-cm^2^-long piece of Whatman No. 1 filter paper in the treatment tube. The flies were exposed to the treatment protocol for one hour. Thereafter, they were transferred to the second tube and monitored for 24 h (Figure 2). Each test was conducted five times, and the concurrent positive control was 1% (*w*/*v*) *α*-cypermethrin and the negative control was distilled water. The adult flies’ antennae, head, thorax, wings, legs, and stomach were monitored to ascertain if they had been knocked down or had died. Knockout or fatality was suggested by a motionless state [24].

We noted and reported the knockdown rate for each formula 1, 5, 10, 15, 30, and 60 min after exposure. We noted the fatality rate 24 h after exposure. The following formula was used to calculate the adult fly knockdown (%K) and mortality (%M) rates [24]:Knockdown rate (%K) = KD/TN × 100(1)Mortality rate (%M) = MT/TN × 100(2)

MT denotes the total number of adult flies killed; the total number of adult flies knocked down is denoted by KD; and TN denotes the total number of adult flies treated.

The following formula was derived for the knockdown index (KI) [26]:KI = KT_50_ of test/KT_50_ of *α*-cypermethrin(3)

KI > 1 indicates that the formula was less harmful than α-cypermethrin; KI < 1 suggests that the formula, either individual or in a binary mixture, was more toxic than *α*-cypermethrin.

The following formula was used to calculate the mortality index (MI) [26]:MI = LT_50_ of test/LT_50_ of *α*-cypermethrin(4)

MI > 1 indicates that the formula was less harmful than α-cypermethrin; MI < 1 suggests that the formula, either alone or in combination, was more toxic than *α*-cypermethrin.

The synergistic mortality index (SMI), which was calculated using the following formula, showed that binary mixtures were more effective than individual formulas at the same concentration [26]:SMI = LT_50_ of the combined formula/LT_50_ of the lone formula(5)

The SMI continuously shows relative synergy; a synergistic effect is demonstrated by SKI < 1, whereas no synergy is indicated by SKI ≥ 1.

The following formula was used to derive the increased knockdown value (IKV), which showed that binary mixtures were more successful at knockdown than single formulas [24]:(6)$$\mathrm{IKV}\;=\;\left[\frac{(KT_{50}\;of\;individual\;formula\;1\;+\;KT_{50}\;of\;individual\;formula\;2)\;-\;KT_{50}\;of\;mixture\;formula\;}{KT_{50}\;of\;individual\;formula\;1\;+\;KT_{50}\;of\;individual\;formula\;2\;+\;KT_{50}\;of\;mixture\;formula}\right]\times 100$$

The following formula was used to derive the increased mortality value (IMV), which showed that binary combinations were more effective than individual formulas [24]:(7)$$\mathrm{IMV}\;=\;\left[\frac{(LT_{50}\;of\;individual\;formula\;1\;+\;LT_{50}\;of\;individual\;formula\;2)\;-\;LT_{50}\;of\;mixture\;formula\;}{LT_{50}\;of\;individual\;formula\;1\;+\;LT_{50}\;of\;individual\;formula\;2\;+\;LT_{50}\;of\;mixture\;formula}\right]\times 100$$

### 2.4. Safety Test on the Dwarf Honeybee (A. florea) and Guppy (P. reticulata) Non-Target Organisms

Using the methodology reported in [39,40], the toxicity of each formula was evaluated against dwarf honeybees (*A. florea*), a non-target pollinator. Sirawut Sittichok, an entomologist, completed scientific identification of the dwarf honeybees after gathering 100 adult workers from the three natural colonies from the KMITL organic farm into an insect box (18 × 25 × 6 cm^3^) and transporting them to our laboratory within an hour. Prior to the topical application test, the workers of the dwarf honeybees were maintained at 27.5 ± 3.0 °C and 78.4 ± 2.0% relative humidity while being fed 50% (*w*/*v*) syrup. For 1:30 min, the dwarf honeybees were sensitized at −8 °C before topical administration. Next, 1 μL of each test formula was applied to their mesonotum. Following application, ten dwarf honeybees were placed in an 8 × 10 × 5 cm^3^ insect cage and fed 50% (*w*/*v*) syrup. Each treatment was administered five times, with 1% (*w*/*v*) *α*-cypermethrin serving as the positive control, and distilled water serving as the negative control. Mortality and abnormal behaviors, including decreased feeding and activity, were noted after 48 h of exposure.

We employed the same method used by Moungthipmalai and Soonwera [31] to test the toxicity of each formula against a non-target aquatic predator guppy (*P. reticulata*). We purchased the guppy fish from an organic farm in Thailand’s Chonburi Province. An animal taxonomist, Tanapoom Moungthipmalai, completed the scientific recognition of the guppy. The guppy was kept separately in a 40 × 60 × 30 cm^3^ plastic box with 5 L of water at 36.0 ± 3.0 °C and 61.0 ± 2.0% relative humidity. Ten adult guppies were placed in a 40 × 28 × 20 cm^3^ plastic bucket with five liters of water to evaluate each formula. Of note, 10,000 ppm of each formula were added as treatment. Each test was conducted five times, using 1% (*w*/*v*) α-cypermethrin and distilled water as positive and negative controls, respectively. Ten days following exposure, data on swimming disability and mortality were gathered.

The following formula was used to calculate the mortality rate of non-target species (%MR) [40]:Mortality rate (%MR) = NT/TT × 100(8)

NT and the total denote the total number of dead adults among the non-target species. TT denotes the number of treated adults.

The following formula was used to derive the safety index (SI) [40]:SI = LT_50_ of non-target species/LT_50_ of *α*-cypermethrin(9)

SI < 1 indicates that the formula was toxic to the non-target species; in comparison, SI > 1 indicates that the formula, either alone or in combination, was non-toxic to the non-target species.

### 2.5. Data Analysis

Each experiment was conducted using a completely randomized design (CRD), and the mean ± SD was used to present the results. To determine the knockdown time KT_50_, the average knockdown data were subjected to a probit regression analysis and LT_50_ values were computed. A one-way analysis of variance (ANOVA) was used to assess the test’s impact. Tukey’s post hoc test (*p* < 0.05) was used to identify significant differences between the control and multiple tested groups if the analysis of variance results demonstrated significant differences between the groups (*p* < 0.05). Version 25 of the SPSS Statistical Software Package was used to perform all statistical analyses.

## 3. Results

### 3.1. Bioassay for Adulticidal Activity

The results presented in Figure 3 illustrate the impact of the adulticide activity on adult fly populations, measured based on KT_50_, KT_90_, and knockdown rate. Achieving 100% knockdown within 60 min, with KT_50_ ranging from 0.06 to 0.12 h and KT_90_ ranging from 0.08 to 0.17 h, six binary mixtures of essential oils (EOs), including geranial + eucalyptol, eucalyptol + *C. citratus*, and geranial + *E. globulus* EO at concentration ratios of 1:1 and 2.5:2.5, together with some individual formulas (1% and 2.5% geranial), were found to be more effective than all other individual EO formulas, which had a KT_50_ ranging from 0.21 to 2.32 h and a KT_90_ ranging from 0.24 to 3.88 h. The binary mixture of 2.5% geranial + 2.5% *E. globulus* EO resulted in the highest knockdown rate, with a KT_50_ of 0.06 h and a KT_90_ of 0.08 h. With an LT_50_ of 0.06 to 0.26 h and an LT_90_ of 0.08 to 0.33 h, all binary mixtures and a few of the individual formulas (1% and 2.5% of geranial) provided a full (100%) mortality effect in 24 h. The lethal time of *α*-cypermethrin was longer than that of the binary mixture of 2.5% geranial + 2.5% *E. globulus* EO (LT_50_ of 0.27 h and LT_90_ of 0.48 h). As expected, and shown in Figure 3, the negative control resulted in no mortality.

The results presented in Figure 4 demonstrate that the six binary mixtures worked more synergistically than all individual EOs. Synergistic mortality index (SMI) values ranged from 0.0004 to 0.0017, values considerably less than one, thus indicating a strong synergy.

All binary mixture formulas displayed in Figure 5 and specific individual formulas (1% and 2.5% of geranial) exhibited KI and MI values that were significantly less than one (KI of 0.44 for 1% geranial and 0.22 for 2.5% geranial + 2.5% *E. globulus*) and (MI of 0.96 for 1% geranial and 0.22 for 2.5% geranial + 2.5% *E. globulus*). Compared to the individual formulas, the efficacies of these binary combinations were higher. The results presented in Figure 6 demonstrate that the IKV and IMV ranged from 92.4% to 96.4% and from 99.7% to 99.9%, respectively. These results suggest that the binary mixture formulas provided more potent adulticidal activity compared to the individual formulas.

### 3.2. Benign Non-Target Bioassay Results

The results presented in Figure 7 demonstrate the level of harm posed by the individual formulas and binary combinations on adult non-target pollinators (*A. florea*) based on mortality, LT_50_ (h), and LT_90_ (h), following exposure for 48 h. Each individual and binary mixture formula exhibited a high level of safety at 48 h (LT_50_ of 49.7 h to 72.5 h, LT_90_ of 58.7 h to 90.3 h, and mortality of 2% to 10%). The formulas all resulted in comparable negative control mortality outcomes (with each one having a 48 h mortality rate of 0%). In contrast, 1% (*w*/*v*) *α*-cypermethrin was highly toxic to adult dwarf honeybees, resulting in 100% mortality rates at 48 h (LT_50_ of 0.20 h and LT_90_ of 0.37 h). The results presented in Figure 8 demonstrate that the safety index (SI) of each combination and the combined EO formula against dwarf honeybees was 248.5 to 362.5 times higher than that of *α*-cypermethrin.

The mortality rate of adult guppies (*P. reticulata*) after 10 days of exposure is displayed in Figure 9, with a mortality rate ranging from 10% to 28%. Each individual formula and binary mixture formula at 10,000 ppm presented less harm to the guppies than 1% *α*-cypermethrin. Furthermore, the negative control exhibited no harmful effects on adult guppies. The most harmful substance for adult guppies, in contrast, was 1% *α*-cypermethrin (100%). As shown in Figure 10, the SI of each formula was 335.5 to 649.4 times greater than that of α-cypermethrin.

## 4. Discussion

Natural pesticides are a safe option for controlling flies and insect vector diseases [7,22,24]. The strong insecticidal and adulticidal actions against flies (*M. domestica*) by both the individual and combined formulas of geranial and *E. globulus* EO have been demonstrated herein. However, the first line of defense against this threat should be adequate management. The use of chemical pesticides is closely linked to ecological harm, public hygiene hazards, and insecticide resistance [41,42]. Due to their widely reported insecticidal action, plant essential oils and their constituents represent an efficient choice for developing novel, potent adulticidal products [43,44]. Notably, the individual and combined geranial and *E. globulus* EO formulas showed great promise. With the lowest KT_50_ and LT_50_ but the highest knockdown mortality, they demonstrated a potent synergy against adult flies in this investigation. The highest level of knockdown and mortality rates, 100%, were observed in the synergistic combination of 2.5% geranial and 2.5% *E. globulus* EO (KT_50_ 0.06 h and LT_50_ 0.06 h). These results are consistent with those reported in other studies [45,46]. The mortality rate of flies was more than 50% for the combined EOs formulas of star anise + geranial, nutmeg + geranial, and α-pinene + geranial [24]. Based on the findings of Moungthipmalai and Soonwera [31], eucalyptol (10.70%), lemongrass EO (100%), and geranial (94.66%) exerted the most substantial adulticidal effects. Geranial is potentially toxic to the neurological system, affecting the insect’s central nervous system [47,48]. The primary neurotoxic mechanisms of EOs against insects include the inhibition of gamma-aminobutyric acid (GABA), which blocks chloride channels and suppresses acetylcholinesterase (AChE), and antagonism with biogenic amine messenger octopamine [49,50]. Acetylcholine levels are regulated by the nervous system enzyme AChE, a minor enzyme that catalyzes its hydrolysis in various species and ultimately causes insect death. Such mechanisms are designed to produce carbamates and organophosphates. However, their use is discouraged [51,52] due to the development of resistance in insects and their established human toxicity. The EOs of lemongrass and eucalyptus induced 100% knockdown and mortality effects against flies [53]. In a similar manner, eucalyptus essential oil has an insecticidal impact on housefly larvae and pupae [54]. In addition, based on the findings of Khan et al. [55], the lemongrass and eucalyptus essential oil mixture has an insecticidal effect on medical pests such as houseflies, bed bugs, and lice. Furthermore, Tennyson et al. [56] demonstrated in their study the impact of eucalyptus EO and citronella on mosquito larva mortality. When used against mill moth larvae, eucalyptus and garlic essential oils (EOs) exhibited neurotoxic effects and reduced acetylcholinesterase (AChE) levels [57]. In addition, thymol and eucalyptol can lower the AChE effect in the larval diamondback moth [58]. Furthermore, in another study, the main component of eucalyptus EO, eucalyptol, inhibited AChE in mosquitoes in a manner comparable to that of organophosphates and carbamates [59]. The beneficial effects of eucalyptol on humans should not be underestimated. Indeed, eucalyptol, which is found in sage essential oil, is the basis for new, modern drugs that improve memory and prevent dementia [60]. In one study, not only did EOs and their primary compounds exert neurotoxic effects in insects, but their vapor also damaged flies’ and mosquitoes’ antennae [61,62]. Primarily, the KI and MI of the combined EOs from geranial and *E. globulus* EO were 0.22, a value considerably lower than one, making it much more potent than the pyrethroid insecticide *α*-cypermethrin. Furthermore, Moungthipmalai and Soonwera [31] demonstrated that 1% geranial was 1.67 times more effective than *α*-cypermethrin as an insecticide against adult flies. From the results presented above, we concluded that the adult flies used in our study have developed resistance to α-cypermethrin. Our findings support the findings of Wang et al. [63], who demonstrated that the flies (*M. domestica*) in their study developed resistance to five insecticides: dichlorvos, propoxur, deltamethrin, *α*-cypermethrin, and permethrin. Furthermore, it was found that the flies’ resistance to some insecticides exhibited instability and might only be reversed after more than a single generation [64,65,66]. However, redistributing adult flies treated with *α*-cypermethrin for 24 breeding sessions increased resistance [67]. In addition, it has been found that *α*-cypermethrin resistance is present in globally prevalent insect pests, such as the German cockroach, the olive fruit fly, the Asian blue tick, the Asian malaria mosquito, and biting flies [67,68,69,70,71].

Insecticide resistance in target insect pests, a highly contaminated environment, and negative health effects are all consequences of chemical pyrethroid insecticide use [72,73]. Insecticides that contain pyrethroid chemicals negatively impact human health, contaminate the environment, and lead to the development of pesticide resistance in their target pests [72,73,74]. As such, in lab settings, adult flies can rapidly develop a very high level of resistance to *α*-cypermethrin. Furthermore, the results of previous studies revealed that field strain flies exhibited 153-fold resistance to *α*-cypermethrin [75]. Similarly, Zhang et al. [76] reported increasing levels of *β*-cypermethrin resistance in their study (14–26%). The 24th generation adult housefly males exhibited *α*-cypermethrin resistant heritability (h^2^) [77]. When a variant is genetically associated with a trait, such as pesticide resistance, a quantitative genetic model is utilized to predict the variation in individual features. The manifestation of a trait is influenced by environmental factors and resistant gene characteristics [78,79]. As more resistant genes are passed down via subsequent generations, a high h^2^ score indicates a significant risk of genetic resistance emerging [80]. High trait variation, as indicated by the declining value of h^2^, can be caused by gene mutation, alternating insecticide application, and natural impacts in the laboratory and surrounding environment [81]. At present, in order to overcome resistance, members of the general public are taking greater risks by using larger doses of insecticides, negatively impacting the environment [82,83]. Synthetic pyrethroid (*β*-cypermethrin) resistance in flies was determined by Zhang et al. [84] to be a hereditary characteristic (single, major, autosomal, and imperfectly recessive trait). The existence of a greater number of resistant flies necessitates more careful and frequent monitoring [85]. The primary attributes of natural essential oils and their principal ingredients are their safety and environmental friendliness. For instance, pollinators, aquatic predators, and earthworms are among the non-target species for which they are frequently deemed safe [27,62,86,87,88,89]. On farmlands, honeybees, as arthropods, are constantly exposed to insecticides [90]. When compared to pyrethroid insecticides (*α*-cypermethrin), all of the individual and binary mixture formulas in the current investigation showed a high SI and LC_50_ and were safer for pollinators (*A. florea*) and aquatic predators (*P. reticulata*). The 2.5% geranial + *2.5% E. globulus* EO showed outstanding innocuousness toward pollinators (*A. florea*), with an LT_50_ of 49.7 h and an SI of 248.5, and aquatic predators (*P. reticulata*), with an LT_50_ of 490.7 h and an SI of 539.3. On the other hand, *α*-cypermethrin was extremely toxic and had the lowest LT_50_ at 0.20 h for *A. florea* and 0.9 h for *P. reticulata*. Similarly to the results of previous studies, Moungthipmalai et al. [91] demonstrated in their study that mollies and guppies could safely consume EOs at 10,000 ppm of cinnamon + geranial, citrus + geranial, and geranial + *trans*-cinnamaldehyde (biosafety index 1.06–2.57). Guppy toxicity was partially induced by eucalyptol (LC_50_ 1701.93–3997.07 ppm) and thymol (LC_50_ 10.99–12.51 ppm) [92]. Furthermore, Sittichok et al. [93] reported in their study that a formula of geranial and *trans*-cinnamaldehyde that was larvicidal and pupicidal for mosquitoes at concentrations of 100–500 ppm had a low biosafety index (0.02–0.12) for guppies; in comparison, a formula of a chemical insecticide at 1 ppm was found to have a high biosafety index (1.0–1.05), indicating that it is harmful for aquatic predators. Furthermore, Soonwera et al. [26] demonstrated in their study that geranial + *trans*-anethole and star anise EO + geranial formulas at doses of 0.5–2% were less harmful to adult guppies and stingless bees than 1% (*w*/*v*) *α*-cypermethrin (100% mortality rate). Compared to α-cypermethrin, these formulas were much safer. The use of primary eucalyptus EO compounds, cineole and eugenol, did not result in honeybee workers’ death [94]. Conversely, after 24–96 h of exposure, the pyrethroid insecticide cypermethrin demonstrated rapid toxicity to stingless bees (LC_50_ 223.69–92.24 ug/mL) [95]. Furthermore, other pyrethroids have been shown to be comparably toxic to non-target species in previous investigations [96,97,98]. Based on the findings of Tome et al. [99], deltamethrin, for instance, had acute toxic effects on some stingless bee species in their study (0.082 ug/bee of *Melipona quadrifasciata* Lepeletier. and 0.014 ug/bee of *Partamona helleri* Friese). In the same vein, stingless bees belonging to the species *Trigona spinipes* Fabricius were acutely harmed by permethrin (LD_50_ 0.07/g/bee) and cypermethrin (LD_50_ 0.070 g/bee) [100]. The stingless bee *Meliponula bocandei* Spinola could therefore serve as an efficient bio-indicator of environmental pesticide contamination because it is extremely sensitive to pyrethroids in the environment [98].

Of particular note, the 2.5% (*v*/*v*) geranial + 2.5% (*v*/*v*) *E. globulus* EO combination exhibited very strong and synergistic adulticide properties. It was also innocuous to non-target species and increased the mortality rate of adult flies (*M. domestica*) to almost 100%. From the above results, it is thus evident that to control housefly populations, this combination should be applied as an effective natural insecticide.

## 5. Conclusions

Our study results provide evidence that a mixture of 2.5% (*v*/*v*) geranial and 2.5% (*v*/*v*) *E. globulus* EO can serve as a reliable synergistic fly adulticidal agent at low concentrations, with very high knockdown rates and mortality indices. This mixture is also safe for non-target species: a primary aquatic predator, guppies, and the primary pollinator, honeybees. Moreover, it was found to be safer and more effective against houseflies than *α*-cypermethrin, a pyrethroid insecticide. From the above results, it is evident that it is poised for further development into a commercially available, environmentally friendly adulticide for managing fly populations and controlling infectious diseases that they vector in agricultural and urban areas.

## Figures and Tables

**Figure 1 insects-16-00855-f001:**
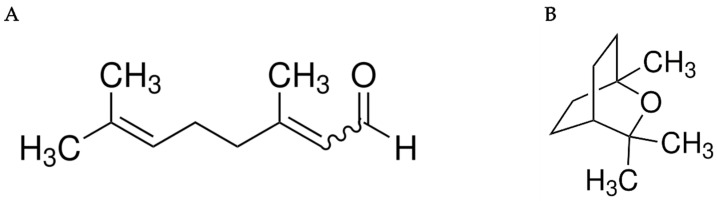
Chemical structure formulas reported by Sigma-Aldrich: geranial (**A**) and eucalyptol (**B**).

**Figure 2 insects-16-00855-f002:**
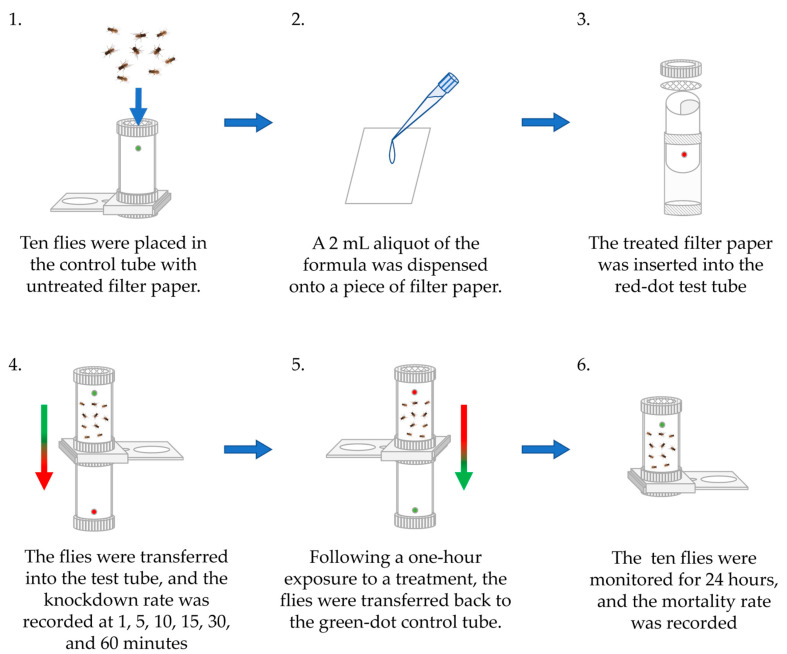
WHO standard susceptibility test of flies’ adulticidal activity.

**Figure 3 insects-16-00855-f003:**
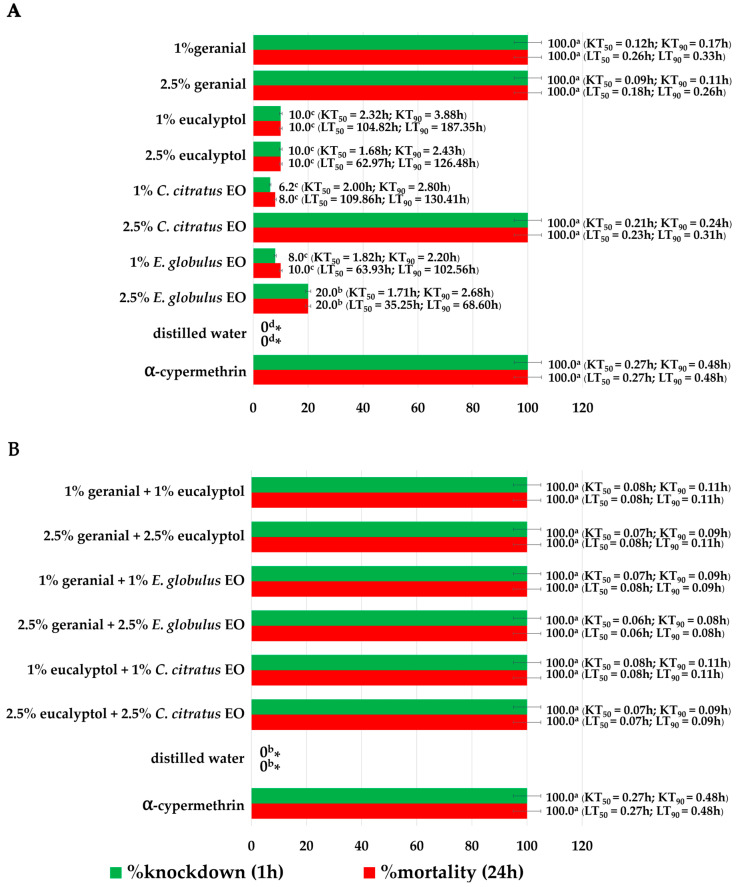
Individual formulas (**A**) and binary mixture formulas (**B**) demonstrated. knockdown; 50% knockdown time (KT_50_) and 90% knockdown time (KT_90_) and mortality; 50% lethal time (LT_50_) and 90% lethal time (LT_90_) against adult flies (*M. domestica*). Note: * Mean knockdown or mortality rates within a column followed by the same letter do not differ significantly (Tukey’s post hoc test *p* < 0.05).

**Figure 4 insects-16-00855-f004:**
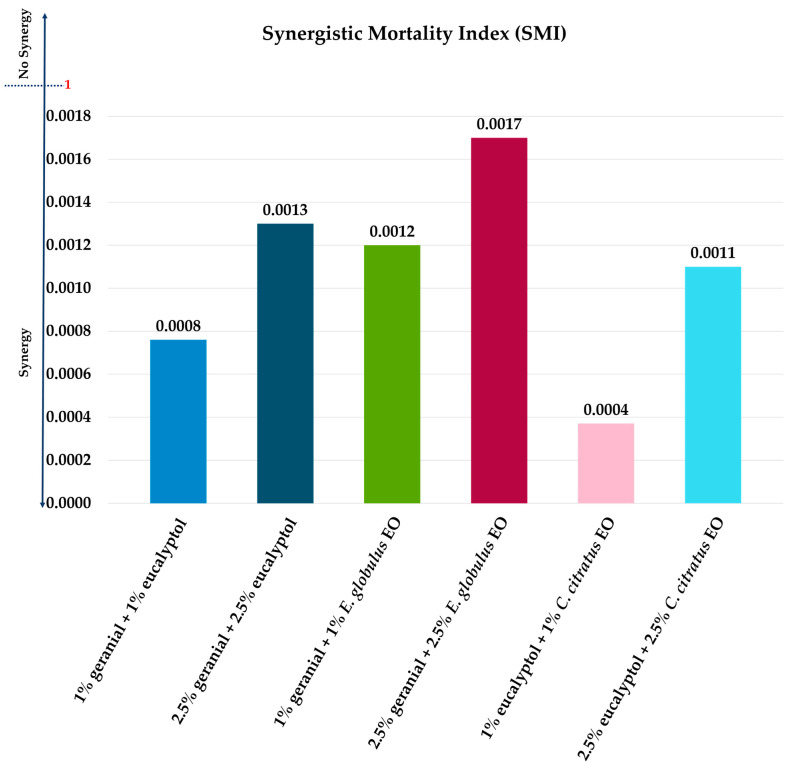
In the above figure, SMI stands for synergistic mortality index against adult flies (*M. domestica*).

**Figure 5 insects-16-00855-f005:**
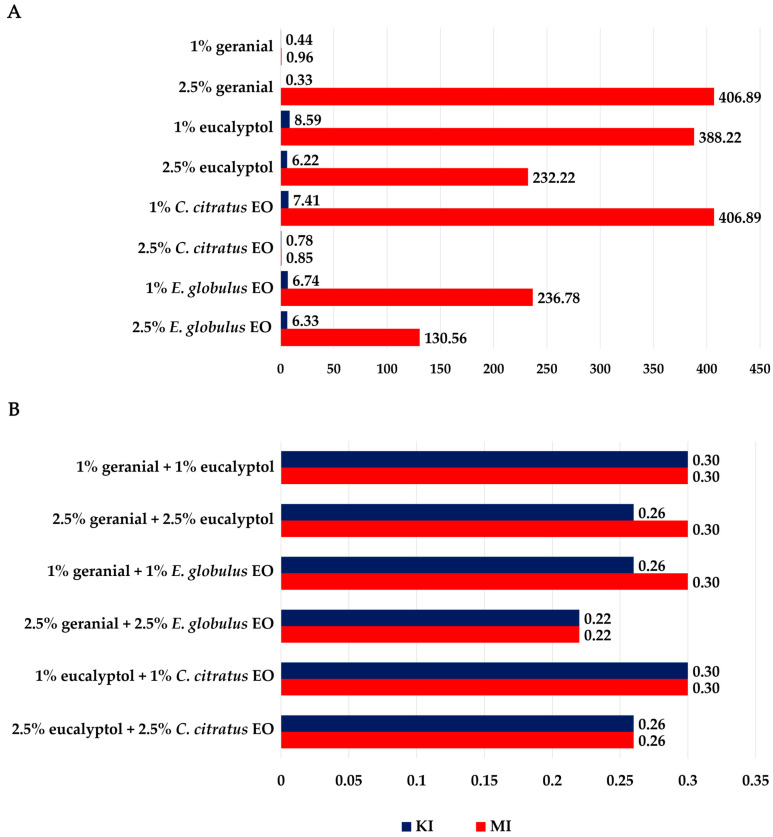
Knockdown index (KI) and mortality index (MI) of individual formulas (**A**) and binary mixture formulas (**B**) against adult flies (*M. domestica*).

**Figure 6 insects-16-00855-f006:**
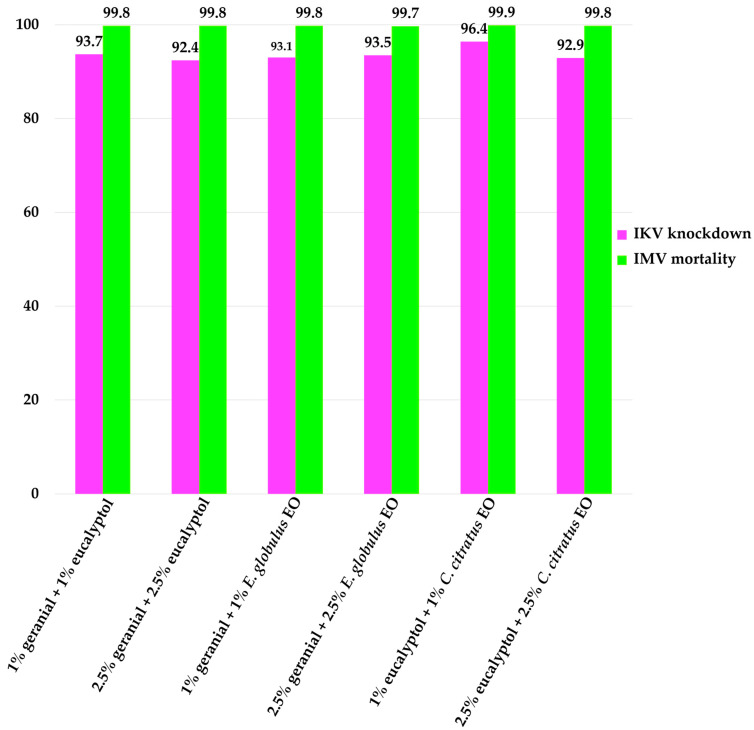
Increasing knockdown value (IKV) and increasing mortality value (IMV) of binary mixture formulas against adult flies (*M. domestica*).

**Figure 7 insects-16-00855-f007:**
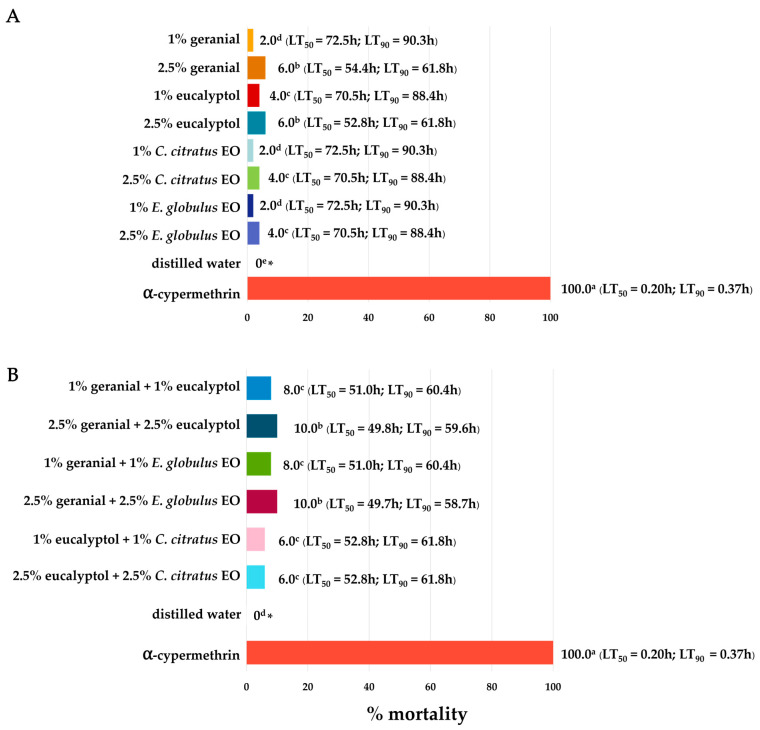
The toxicity of individual formulas (**A**) and binary mixture formulas (**B**) to adult dwarf honeybees (*A. florea*). Note: * Mean mortality rates within a column followed by the same letter do not differ significantly (Tukey’s post hoc test *p* < 0.05).

**Figure 8 insects-16-00855-f008:**
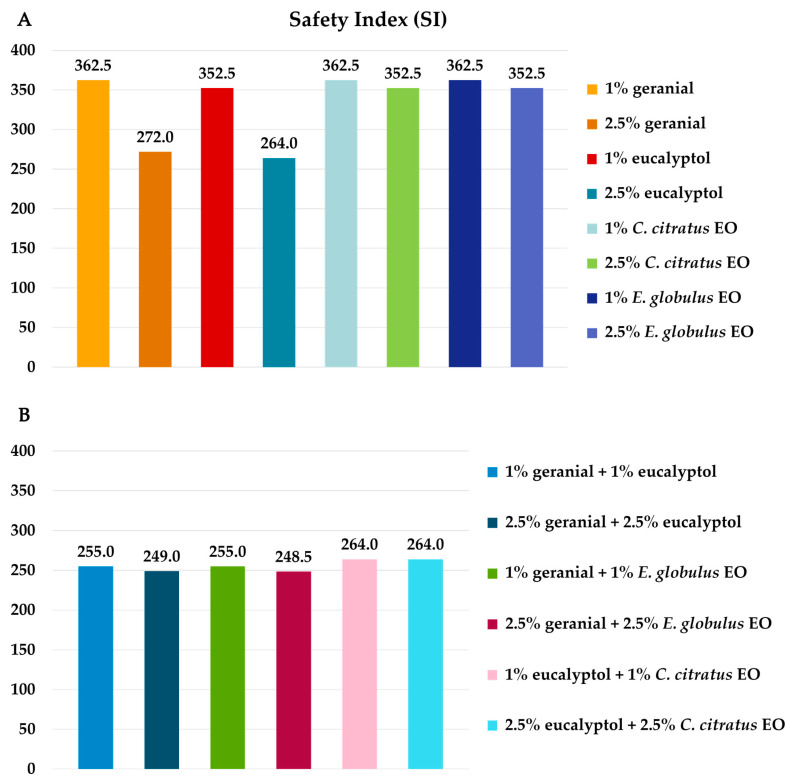
Safety index (SI) of each individual formula (**A**) and binary mixture formula (**B**) compared with that of α-cypermethrin against the dwarf honeybee (*A. florea*).

**Figure 9 insects-16-00855-f009:**
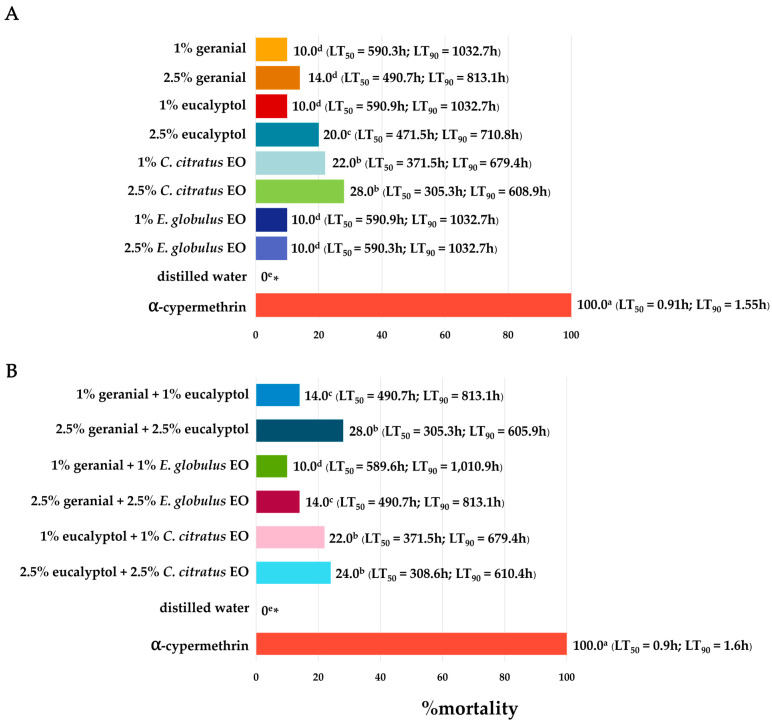
Mortality, LT_50_, and LT_90_ of individual formulas (**A**) and binary mixture formulas (**B**) used on adult guppies (*P. reticulata*) for 10 days of exposure. Note: * Mean mortality rates within a column followed by the same letter do not differ significantly (Tukey’s post hoc test *p* < 0.05).

**Figure 10 insects-16-00855-f010:**
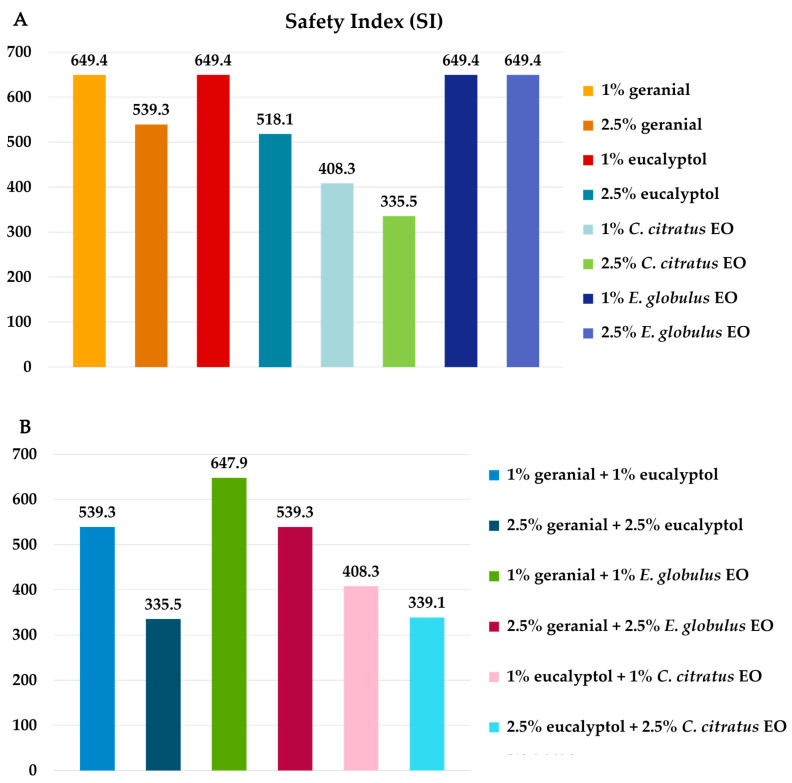
Safety index (SI) of each individual formula (**A**) and binary mixture formula (**B**) compared with that of α-cypermethrin against guppies (*P. reticulata*).

**Table 1 insects-16-00855-t001:** EOs and their primary chemical constituents used against several organisms.

EOs and Their Main Chemical Compound	Against Organisms	Action	Ref.
Geranial	*M. domestica*	Inhibited acetylcholinesterase activity	[24]
*Trans*-anethole	*M. domestica*	Induced cell membrane damage	[24]
*Trans*-cinnamaldehyde	*M. domestica* and *Aede aegypti*	Inhibited cytokinesis activity and excitatory to the neurons	[29,30]
Eucalyptol	*M. domestica*	Against the octopamine receptor and acetylcholine	[31]
α-pinene	*M. domestica*	Inhibited AChE activity	[7]
*Illicium verum* Hook. f. EO	*M. domestica* and *Ae. Aegypti*	Caused nervous toxicity	[24,32]
*E. globulus* EO	*M. domestica*	Inhibited AChE activity	[31]
*C. citratus* EO	*M. domestica* and *Ae. Aegypti*	Inhibited AChE activity and the toxicity of the olfactory	[25,32]

**Table 2 insects-16-00855-t002:** Combined EO formulas against adult flies.

EO Constituents and Their Combinations	Formulas
Geranial	1% geranial in 70% ethyl alcohol
	2.5% geranial in 70% ethyl alcohol
Eucalyptol	1% eucalyptol in 70% ethyl alcohol
	2.5% eucalyptol in 70% ethyl alcohol
*C. citratus* EO	1% *C. citratus* EO in 70% ethyl alcohol
	2.5% *C. citratus* EO in 70% ethyl alcohol
*E. globulus* EO	1% *E. globulus* EO in 70% ethyl alcohol
	2.5% *E. globulus* EO in 70% ethyl alcohol
Geranial + eucalyptol	1% geranial + 1% eucalyptol in 70% ethyl alcohol
	2.5% geranial + 2.5% eucalyptol in 70% ethyl alcohol
Geranial + *E. globulus* EO	1% geranial + 1% *E. globulus* EO in 70% ethyl alcohol
	2.5% geranial + 2.5% *E. globulus* EO in 70% ethyl alcohol
Eucalyptol + *C. citratus* EO	1% eucalyptol + 1% *C. citratus* EO in 70% ethyl alcohol
	2.5% eucalyptol + 2.5% *C. citratus* EO + in 70% ethyl alcohol

## Data Availability

The original contributions presented in this study are included in the article. Further inquiries can be directed to the corresponding author.

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
