# Peer review of "Binary Mixtures of Essential Oils: Potent Housefly Adulticides That Are Safe Against Non-Target Species"

_insects, 2025, doi:10.3390/insects16080855_

Round 1

Reviewer 1 Report

Comments and Suggestions for Authors

This manuscript investigates the toxic effects of various essential oils and their main components on house flies, as well as their non-target effects on other organisms. It is a well-designed and well-prepared study; however, a thorough revision in terms of language and expression would help improve the overall clarity and quality of the manuscript.

Some comments; 

  1. All species and genus names should be italicized throughout the manuscript. The full scientific name with author citation (Musca domestica L,) must be given at first mention, and abbreviated properly (e.g., M. domestica) in subsequent uses. Italics must be used consistently for all Latin genus/species names.
  2. The text mentions mixtures such as 1:1 and 2.5:2.5, which appear to represent combinations of 1% and 2.5% solutions. This should be clearly explained in both the Materials and Methods and simple summary sections.
  3. Arranging the keywords in alphabetical order may enhance readability and provide greater ease for the readers.
  4. All genus and species names in the text should be italicized, such as Salmonella, Shigella, and Giardia lamblia.
  5. The English in the Introduction section appears to be somewhat limited, with frequent repetition of the same words, which makes the narrative harder to follow. For example, terms like 'synthetic pesticides' and 'natural insecticides' are used repeatedly without variation.
  6. The compound name α-cypermethrin appears inconsistently throughout the text, with some instances written as 'αcypermethrin'. A uniform spelling should be used consistently across the manuscript.
  7. The formatting of references is inconsistent. Journal names should be abbreviated according to indexing standards (e.g., PubMed, ISO), and all genus/species names should be italicized. Punctuation must also be standardized.
  8. Line 288: “at 10,000 presented” should be corrected to “at 10000 ppm”.
  9. The terms “mg/L” and “ppm” are equivalent; therefore, only one unit should be used consistently throughout the manuscript.

Comments on the Quality of English Language

I suggest that the manuscript be carefully reviewed for language and clarity to enhance its readability.

Reviewer 2 Report

Comments and Suggestions for Authors

Dear Authors,

Your study on the use of essential oil mixture as insecticides is of great interest, given that the use of conventional insecticides has multiple adverse impacts on the environment and human health.

The scientific methods used are clearly and exhaustively described as the presentation of the results and their discussion.

However, I would like to suggest some improvements that I believe are necessary to enhance the quality of the manuscript.

Comments on the Quality of English Language

In some cases, I believe that the English language could be improved to facilitate a better understanding of scientific research.

Reviewer 3 Report

Comments and Suggestions for Authors

This aim of this paper is to investigate the insecticidal potential of two essential oils (EOs) (Eucalyptus globulus and Cymbopogon citratus), alone and in synergy with two primary active compounds (geranial and eucalyptol). 

While the paper is well-written, there are some main issues that need to be resolved before I believe this paper can be considered for publication. These issues arise mainly in the Materials and Methods section of this paper.

1) Why did the authors use a 70% (v/v) ethanolic stock solution? 

2) How does 1% and 2.5% of the EOs compare to the concentration found in nature in the plants in question?

3) Did the authors ever try an only alcoholic solution as a control? If so, this was not mentioned.

4) Why did the authors not provide a schematic of the experimental protocol used to perform the "flies' adulticidal activities" testing? As written, it was nearly impossible to understand how testing was carried out with test tubes. Also, how were the flies tested in the "testing" and "untesting" section of the tubes? What does this mean? As a result of the testing protocol being poorly explained, it was difficult to judge the effectiveness and results obtained from the experiments being carried out.

5) Why did the authors use timeframes for testing of one hour versus 24 hours? Why specifically these time periods? This was not explained.

6) If the experimental protocol was adapted from another published paper, I was unable to find the reference from those provided by the authors. In addition, information pertaining to Ref. 38 (p. 4) (re: WHO) was unretrievable as the link provided by the authors was dead.

7) Some figures in the Results section (i.e., Figs. 2, 6, 8) were very difficult, if not impossible, to read as the chosen font was too small and verifiability of the statistics performed, as well as whether the results were significant or not.

8) Having stated the above-mentioned comments, I was unable to determine if all of the results in this paper were accurate and would recommend that the authors address these major concerns.
